# Numerical Conversion Method for the Dynamic Storage Modulus and Relaxation Modulus of Hydroxy-Terminated Polybutadiene (HTPB) Propellants

**DOI:** 10.3390/polym15010003

**Published:** 2022-12-20

**Authors:** Yongchao Ji, Liang Cao, Zhuo Li, Guoqing Chen, Peng Cao, Tong Liu

**Affiliations:** 1College of Science, Inner Mongolia University of Technology, Hohhot 010000, China; 2College of Architecture and Civil Engineering, Beijing University of Technology, Beijing 100124, China; 3School of Architecture and Engineering, Northeast Electric Power University, Jilin 132012, China; 4No. 41 Institute of the Sixth Academy of China Aerospace Science and Industry Corporation, Hohhot 010010, China

**Keywords:** dynamic storage modulus, relaxation modulus, hydroxy-terminated polybutadiene (HTPB) propellant, viscoelastic, constitutive model

## Abstract

As a typical viscoelastic material, solid propellants have a large difference in mechanical properties under static and dynamic loading. This variability is manifested in the difference in values of the relaxation modulus and dynamic modulus, which serve as the entry point for studying the dynamic and static mechanical properties of propellants. The relaxation modulus and dynamic modulus have a clear integral relationship in theory, but their consistency in engineering practice has never been verified. In this paper, by introducing the “catch-up factor *λ*” and “waiting factor *γ*”, a method for the inter-conversion of the dynamic storage modulus and relaxation modulus of HTPB propellant is established, and the consistency between them is verified. The results show that the time region of the calculated conversion values of the relaxation modulus obtained by this method covers 10^−8^–10^4^ s, spanning twelve orders of magnitude. Compared to that of the relaxation modulus (10^−4^–10^4^ s, spanning eight orders of magnitude), an expansion of four orders of magnitude is achieved. This enhances the expression ability of the relaxation modulus on the mechanical properties of the propellant. Furthermore, when the conversion method is applied to the dynamic–static modulus conversion of the other two HTPB propellants, the results show that the correlation coefficient between the calculated and measured conversion values is *R*^2^ > 0.933. This proves the applicability of this method to the dynamic–static modulus conversion of other types of HTPB propellants. It was also found that *λ* and *γ* have the same universal optimal value for different HTPB propellants. As a bridge for static and dynamic modulus conversion, this method greatly expands the expression ability of the relaxation modulus and dynamic storage modulus on the mechanical properties of the HTPB propellant, which is of great significance in the research into the mechanical properties of the propellant.

## 1. Introduction

Mankind has been passionate about space exploration since ancient times. The development of spacecraft has made space exploration and even space travel possible, and the key challenge in realizing this vision is the power source of the spacecraft [1]. With the development of modern technology, high-performance spacecraft have attracted much attention [2]. As the power source of a rocket or spacecraft, the chemical and mechanical properties of propellants determine the height, range, and service life of the spacecraft [3,4,5]. Propellants are typically viscoelastic materials and generally go through processes such as curing and cooling, long-term storage, carrier mobility, and ignition and launch [6,7]. During spacecraft engine transportation and flight, propellant grain is subjected to a variety of complex dynamic loads such as shock, vibration, acceleration, thermal stress, ignition pressure, etc., [8]; while during the storage process, due to the long-term static loading of gravity, propellant grain also exhibits relaxation and creep characteristics [9]. As a result of the abovementioned loads, propellant grains will suffer from fatigue, aging failure, crack, and other behaviors which will seriously impair the survival ability and combat ability of solid engines [10,11,12]. Therefore, an in-depth dynamic and static mechanical property study are needed for solid propellant grain, and this is of great importance for the proper operation of the solid engine.

As the key point in studying the mechanical properties of viscoelastic materials, the relaxation modulus and dynamic modulus have a wide range of applications in the overall design, safety inspection, structural integrity assessment, and simulation calculation of the solid engine [13,14]. They are not actually independent and unrelated, rather they have a clear integral conversion relationship in theory, but the limitation of the extremely accurate theoretical relationship means that their application in the engineering field will be limited [15]. However, the limitations of extremely accurate theoretical relationships mean that their application in engineering is significantly limited [16]. It is difficult to convert the relaxation modulus and dynamic modulus in practical engineering, which creates a large inconsistency in their application in engineering practice and the finite element, resulting in the problem that relevant dynamic and static studies cannot achieve mutual verification. Therefore, many scholars have conducted relevant studies on this problem [17,18,19,20,21,22,23,24]. Zhao and Shen et al. proposed an approximate conversion equation for calculating the stress relaxation modulus of solid propellants via the derivation, analysis, and correction of the theoretical conversion equation of dynamic–static mechanical properties based on dynamic–static viscoelastic experiments [25]. Zhang et al. investigated dynamic mechanical behaviors of the HTPB propellant in the strain rate range of 10^3^–10^4^ s^−1^ using a Split-Hopkinson pressure bar (SHPB) device and captured the deformation and fracture development of the propellant in real-time using a high-speed digital camera and the SHPB device. The obtained stress–strain curves showed that mechanical characteristics such as ultimate stress and strain energy were strongly dependent upon the strain rate [26]. Li and Wu et al. studied viscoelastic parameters of the open-graded friction course (OGFC) under dynamic–static load and conducted uniaxial compression creep tests and dynamic modulus tests to obtain the creep compliance of OGFC. They also derived the relaxation modulus and dynamic modulus of OGFC by viscoelastic theory and established the functional relationship of viscoelastic parameters of OGFC under dynamic and static load [27]. Yang and Peng et al. proposed a method for calculating the modified stress relaxation modulus of solid propellants using the master curve of dynamic storage modulus based on the relationship between the dynamic storage modulus and stress relaxation modulus in one-dimensional linear viscoelastic theory. Specifically, a derivation of the theoretical conversion was carried out first and followed by the proportional correction. However, this method is complicated and the physical meaning of the proportional correction is unclear [28]. Moreover, it has been demonstrated that the proposed dynamic–static conversion equations based on polymers such as plastics and rubbers do not apply to solid propellants. Further, current studies mainly focus on the strength and stiffness of materials, the analytical prediction methods, and the development of constitutive models [29,30,31]. However, materials in industrial applications are subjected to not only static but also dynamic loads, probably both at the same time, or both occurring sequentially [32]. Therefore, it is of great importance to study the combination of the dynamic and static studies of propellants to achieve the inter-conversion of the relaxation modulus and dynamic modulus [33,34], so that dynamic and static experiments or simulations can mutually corroborate. So, how can the numerical conversion of the dynamic storage modulus and relaxation modulus in engineering applications be realized? we have researched this issue.

In this paper, the relaxation modulus and dynamic storage modulus are studied at the same frequency or timescale by mathematical transformation and their curves show the same change trend (monotonic increasing or decreasing). It can be described as the relationship between “catching up” and “waiting”. Therefore, a simple method for converting the dynamic storage modulus and relaxation modulus is proposed by introducing the “catch-up factor λ” and “waiting factor γ” based on the basic linear viscoelastic theory and Boltzmann superposition principle. Then, by analyzing the relationship between λ and γ, this method is shown to explore the inherent properties of HTPB propellants. Finally, the applicability of this method to HTPB propellants and its practical significance in simulation is illustrated by the applicability verification calculation and finite element simulation.

## 2. Materials and Experiments

### 2.1. Material Component and Sample Preparation

Three propellants are used in this paper, namely HTPB-A, HTPB-B, and HTPB-C. Their main chemical components include ammonia perchlorate, aluminum powder, end-hydroxy polybutadiene, curing agent, etc., [35]. The respective component content of the three propellants is shown in Figure 1.

Due to a certain danger in the preparation process of propellants [36], HTPB-A and HTPB-B were provided by the cooperative. HTPB-C is made with the equivalent content of Sodium Chloride instead of Ammonium Perchlorate. This removes the deflagration properties from the preparation process and prepared product of HTPB-C and maintains the mechanical properties similar to the real propellant while ensuring safety. Therefore, HTPB-C was prepared in the lab, and the preparation process is shown in Figure 2. All three prepared propellant billets were cut into rectangular samples of 30 ± 1 mm × 10 ± 0.2 mm × 5 ± 0.2 mm for testing (Figure 2).

### 2.2. Experimental Conditions

A dynamic thermomechanical analyzer (DMA) is an effective mechanical analysis tool used to determine the mechanical properties of materials under time, temperature, and multiple combinations of conditions [37]. In this experiment, the DMA was used to test the relaxation modulus and dynamic modulus of three propellant samples, under the experimental conditions listed in Table 1.

The relaxation modulus test was conducted by stretching the sample to 5% strain in a short time and monitoring the stress change within 1800 s. For the dynamic modulus test, it was conducted by stretching the sample to 5% strain, apart from that, dynamically loading it at 1% strain with a sweep frequency range of 1–200 Hz. Due to the different test temperatures involved in this experiment [38], three repetitions under each temperature were conducted to reduce the error, so a total of 102 sets of tests were performed. Experimental data were analyzed according to the time–temperature equivalent principle and GJB770B-2005 [39].

## 3. Results and Analysis

### 3.1. Experimental Results

The experimental results of the dynamic modulus and relaxation modulus, and related data are shown in Figure 3; Figure 3a–d are the dynamic modulus curves, storage modulus curves, loss modulus curves, and loss factor curves for HTPB-A at five different temperatures, respectively; Figure 3e shows the loss factor master curve for HTPB-A; Figure 3f shows the dynamic modulus offset factor curve for HTPB-A; Figure 3g shows the relaxation modulus curves for HTPB-A at six different temperatures; Figure 3h shows the master curves of the storage modulus and relaxation modulus for HTPB-A; Figure 3i–l show the master curves of the relaxation modulus, dynamic modulus, storage modulus and loss modulus of three propellants, respectively. 

From the analysis of the obtained experimental curves, it is shown that the dynamic modulus, storage modulus, and loss modulus are positively correlated with load frequency; the growth rate of the dynamic modulus and storage modulus first increases with frequency and then decreases slowly; the growth rate of the loss modulus increases continuously with frequency, as shown in Figure 3a–c. Further, below 0 °C, the reduced temperature has a significant effect on the increase in the modulus of the propellants; above room temperature, however, the increase in temperature has a very limited effect on the softening of the propellants. As shown in Figure 3d, at 0–200Hz, the temperature change has an obvious effect on the loss factor. Above 25 °C, the loss factor tends to increase and then decrease with an increase in frequency, while below −20 °C, the loss factor shows an opposite trend. It can be concluded that there is a temperature interval between −20 °C and 25 °C where the loss factor remains virtually unchanged with an increase in frequency. As shown in Figure 3e, the loss factor master curve exhibits the trend of a full cycle “sine wave” in the frequency range of 10^−2^–10^5^ Hz. Figure 3h shows that the modulus of the same HTPB propellant under dynamic load is significantly higher than the static load [40]. Comparing the relaxation modulus of the three propellants, as shown in Figure 3i, it can be observed that the relaxation modulus of HTPB-A and HTPB-C at different moments are closer, but both are lower than that of HTPB-B. The dynamic modulus and storage modulus of the three propellants are obviously different, but the loss modulus is closer (shown in Figure 3i–l). This might indicate that the dynamic mechanical properties of HTPB propellants are mainly determined by the storage modulus.

### 3.2. Theoretical Analysis of the Dynamic–static Modulus Conversion Method

First, the relaxation modulus and dynamic modulus are expressed as follows:(1)$$E(t)=E_{\infty }+E_{0}\phi (t),$$
(2)$$E(\omega )=E^{\prime }\left(\omega \right)+E^{\prime \prime }\left(\omega \right).$$

According to viscoelastic mechanics, the theoretical conversion relationship of the storage modulus $E^{\prime }(\omega )$, loss modulus $E^{\prime \prime }(\omega )$, and relaxation modulus is E(t), as follows [41]:(3)$$E^{\prime }\left(\omega \right)=E_{\infty }+\omega E_{0}\int_{0}^{\infty }\phi (t)\sin\omega tdt,$$
(4)$$E^{\prime \prime }\left(\omega \right)=\omega E_{0}\int_{0}^{\infty }\phi (t)\cos\omega tdt.$$where $E_{\infty }$ is the equilibrium modulus, ϕ(t) is the relaxation function, and ω is the angular frequency.

If $E^{\prime }(\omega )$ and $E^{\prime \prime }(\omega )$ are known, E(t) can be obtained from the Fourier reverse conversion of Equations (1) and (2).
(5)$$E(t)=E_{\infty }+\frac{2}{\pi }\int_{0}^{\infty }\frac{E^{\prime }(\omega )-E_{\infty }}{\omega }\sin\omega tdt,$$
(6)$$E(t)=E_{\infty }+\frac{2}{\pi }\int_{0}^{\infty }\frac{E^{\prime \prime }(\omega )}{\omega }\cos\omega tdt.$$

Because $E_{\infty }$ is not easy to measure in engineering practice, and the above equations are all infinite integrals that require a long relaxation observation time in the test, or a quite wide dynamic sweep range. This makes it difficult to apply the theoretical conversion equation in engineering practice [42].

### 3.3. Introduction of the “Catch-up Factor λ” and “Waiting Factor γ”

Knowing the difficulty in using theoretical equations, there are some approximate dynamic–static modulus conversion equations that have been generated for engineering purposes, for example [15]:(7)$$E(t)\approx E^{\prime }(\omega )\left|{}_{t=\frac{1}{\omega }}\right.,E^{\prime }(\omega )\approx E(t)\left|{}_{\omega =\frac{1}{t}}\right.,$$
(8)$$E^{\prime }\left(\omega \right)\approx E(t)+0.86\left[E(t)-E(2t)\right]\left|{}_{\omega =\frac{1}{t}}\right.,$$
(9)$$E(t)\approx E^{\prime }\left(\omega \right)\left|{}_{t=\frac{2}{\pi \omega }}\right.,E^{\prime }\left(\omega \right)\approx E(t)\left|{}_{\omega =\frac{2}{\pi t}}\right.,$$
(10)$$E^{\prime \prime }\left(\omega \right)\approx -0.47\left[E(2t)-E(4t)\right]+1.674\left[E(t)-E(2t)\right]+0.198\left[E(0.5t)-E(t)\right]\left|{}_{\omega =\frac{1}{t}}\right..$$

The above conversion Equations (7)–(10) are either too complex or not accurate, and most are summarized by the dynamic–static properties of polymers such as plastic and rubber, which are not suitable for propellant modulus conversion. 

However, these studies also bring inspiration. As we all know, for dynamic load, there will always be $\omega =2\pi f,f=\frac{1}{T}$, where f is the frequency, and T is the dynamic loading cycle. Transforming ω for f in Equation (7), the following equations are obtained:(11)$$E(t)\approx E^{\prime }\left(f\right)\left|{}_{t=\frac{1}{\pi^{2}f}}\right.,$$
(12)$$E(t)\approx E^{\prime }\left(T\right)\left|{}_{t=\frac{T}{\pi^{2}}}\right..$$

Therefore, when the quantitative mathematical relationship between the dynamic loading recycles and relaxation time is established, the approximate conversion of the dynamic–static modulus can be achieved. This can also be reasonably explained in engineering practice when T→∞ a dynamic process is equivalent to a quasi–static process. In other words, a dynamic process with an infinitely long cycle is equivalent to a static one, and a static process with extremely short monitoring time is equivalent to a dynamic one. The two are similar to trains traveling on two parallel lines but at different speeds, although the head and tail overlap sometimes, it is extremely difficult to move forward in parallel. So, what will happen if one waits and another catches up? This prompts the introduction of λ and γ.

Because both the relaxation modulus and the dynamic storage modulus can be fitted with the Prony series, and the core of the Prony series is the exponential function, so λ and γ are also introduced in the exponential form. The relaxation modulus E(t) and storage modulus $E^{\prime }(\omega )$ can be expressed as:(13)$$E(t)\approx E^{\prime }\left(T\right)\left|{}_{t=(\frac{T}{\pi^{2}}{)}^{\lambda }}\right.,E(t)\approx E^{\prime }\left(f\right)\left|{}_{t=(\frac{1}{\pi^{2}f}{)}^{\lambda }}\right.,$$
(14)$$E(t)\approx E^{\prime }\left(\omega \right)\left|{}_{t={\left(\frac{2}{\pi \omega }\right)}^{\lambda }}\right.,E^{\prime }\left(\omega \right)\approx E(t)\left|{}_{\omega =2\pi f=2\pi {\left(\frac{1}{\pi^{2}t}\right)}^{\gamma }}\right..$$

Although E(t) and $E^{\prime }\left(T\right)$ in function form are monotonically decreasing with time, the initial value $E^{\prime }\left(T\right)$ is much higher than that of E(t). Therefore, to be consistent with E(t), $E^{\prime }\left(T\right)$ needs to accelerate the decreasing trend to “catch up”; on the contrary, to be consistent with $E^{\prime }\left(T\right)$, E(t) needs to delay the decreasing trend to “wait”. In this way, with a clear physical meaning and a simple conversion form, the dynamic–static modulus conversion equation (Equation (14)) can be obtained.

### 3.4. Specific Values Determination of the “Catch-Up Factor λ” and “Waiting Factor γ”

To determine the specific values of λ and γ, the coefficient of determination (*R*^2^) was used as an evaluation index to characterize the degree of fit between calculated and measured conversion values.$R^{2}$, known as the correlation coefficient or goodness of fit, is a statistical term that measures the gap between the expected values of the model and actual values obtained in reality [43]. It is often used to evaluate the fit effect of a model on observed values, or evaluate the degree of fit between predicted and measured values [44]. The calculation equation of $R^{2}$ can be expressed as:(15)$$R^{2}=\frac{\sum_{i=1}^{n}{\left(Y_{i}-\bar{X}\right)}^{2}}{\sum_{i=1}^{n}{\left(X_{i}-\bar{X}\right)}^{2}}=1-\frac{\sum_{i=1}^{n}{\left(X_{i}-Y_{i}\right)}^{2}}{\sum_{i=1}^{n}{\left(X_{i}-\bar{X}\right)}^{2}}.$$
where $X_{i}$ is the measured value, $Y_{i}$ is the calculated conversion value, $\bar{X}$ is the measured average value, and $\bar{Y}$ is the calculated average conversion value. The value range of *R*^2^ is 0 to 1, and the closer to 1, the better the predicted values of measured fit values will be.

After the calculation using Equation (14), the measured relaxation modulus and dynamic storage modulus of HTPB-A propellant were converted. Finally, the results show that when λ=1.445, the result of $R^{2}$ between the calculated relaxation modulus from the dynamic storage modulus and the observed relaxation modulus values is $R^{2}$ ≥ 0.995; when γ=0.692, the result of *R*^2^ between the calculated dynamic storage modulus from the relaxation modulus and the observed dynamic storage modulus values is $R^{2}$ ≥ 0.989, as shown in Figure 4 and Figure 5.

Figure 4 shows that the time range of E(t) conversion calculation values expands about four orders in magnitude compared to that of the measured values. This is significant for the study of propellant because, in the real relaxation modulus experiments, the initial strain loading always takes a certain time (about 2 s) due to the limit of experimental equipment, and during this time, stress relaxation has been occurring until the initial strain reaches the set value. Thus, the closer to the moment of zero load, the more difficult it is to collect the relaxation data, which makes the initial value of the relaxation modulus obtained from the experiment much lower than the actual modulus value at the moment of zero load. 

Using the time–temperature equivalent principle, it is hoped that by lowering the temperature, the relaxation modulus of the propellant can be increased so that the modulus data close to the moment of zero load as much as possible through a reasonable shift operation can be obtained [45]. Since it is impossible to lower the temperature indefinitely to increase the relaxation modulus, the time–temperature equivalent principle is limited to the extension of the relaxation modulus master curve. However, the modulus conversion method mentioned above has an obvious effect on the extension of the relaxation modulus master curve. Specifically, this method can extend the relaxation modulus master curve by four orders of magnitude at once in the direction close to the moment of zero load. (L^m+^ mentioned below is obtained by extending the relaxed modulus main curve using this modulus conversion method).

Similarly, the experiment with a dynamic loading frequency of less than 1 Hz is not only time-consuming but it is also difficult to achieve a vibration frequency close to 0 Hz. The conversion value of the dynamic storage modulus calculated from the relaxation modulus can also be expanded in the direction of frequency close to 0 Hz, as shown in Figure 5.

In summary, the dynamic and static experiments can exactly compensate for each other through the inter-conversion of the relaxation modulus and dynamic storage modulus. Generally, static experiments are simple and easy to perform when t≥1s, while dynamic experiments are more suitable when t<1s [15]. Thus, using this modulus conversion method, combining static and dynamic experiments with suitable experimental equipment and temperature conditions makes it possible to characterize the mechanical properties of propellants over a time range of a dozen orders of magnitude from 10^−8^ s to 10^8^ s.

In addition, it is common that there is a special reciprocal relationship among viscoelastic physical quantities in viscoelastic theory. For example, the relaxation modulus and creep compliance result in 1, in the case of convolution (Equation (16)) [46]. In this paper, using the modulus conversion method, it is found that λ and γ also have a perfect reciprocal relationship. Combined with the following Equation (16), it shows that this method is the excavation of the inherent properties of HTPB propellants, but this requires the verification of a large amount of experimental data.
(16)$$\left\{\begin{array}{c} \int_{0}^{t}E(t-\tau )dF(\tau )=\int_{0}^{t}F(t-\tau )dE(\tau )=1\\ or\\ E^{*}(\omega )D^{*}(\omega )=1. \end{array}\right.$$
where E(t) is the relaxation modulus, F(t) is the creep compliance, t is the observation time, τ is the integral variable, $E^{*}(\omega )$ is the complex modulus, and $D^{*}(\omega )$ is the complex compliance.

## 4. Applicability Verification

To verify the applicability of the static and dynamic modulus conversion method with the “catch-up factor λ” and “waiting factor γ” to HTPB propellants with different material component contents, Equation (14) was used to convert the dynamic–static modulus of HTPB-B and HTPB-C, respectively, and the results are shown in Figure 6.

In Figure 6, the gray anastomosis area is the area where conversion calculation values and observed values coincide. It is the embodiment of conversion calculation values and observed values containing the same mechanical information. The pink area is the expansion area and it reflects that the mechanical information of conversion calculation values is more than that of observed values. As shown in Figure 6, HTPB-A and HTPB-C have little difference in performance, and their initial modulus differs significantly from that of HTPB-B, by approximately 10 times. However, all three apply to Equation (10) with respect to λ and γ, which reflects the good applicability of this conversion method for HTPB propellants. As shown in Figure 4, Figure 5 and Figure 6, the size of the expansion region of the markers varies, which is caused by the different experimental conditions. To obtain a wider expansion region in the conversion of the dynamic storage modulus to the relaxation modulus, the dynamic experiment temperature needs to be reduced or the loading frequency needs to be increased. Similarly, to obtain a wider expansion region in the conversion of the relaxation modulus to the dynamic storage modulus, the static experiment temperature needs to be increased or the relaxation observation time extended.

In particular, in the applicability verification calculations for HTPB-A, HTPB-B, and HTPB-C, λ = 1.445 and γ = 0.692 were used and the *R*^2^ result between the calculated conversion values and the measured values was *R*^2^ ≥ 0.933. There is already a high degree of fit, however, to obtain a higher value of *R*^2^, then λ and γ need to be fine-tuned. After the fine-tuning, the results for λ, γ, and *R*^2^ can be seen in Table 2.

By analyzing the results, Table 2 shows that for HTPB propellants, there should exist a fixed constant value for both the λ and γ used for static–dynamic modulus conversion, so that the calculated and measured values of the modulus conversion coincide. This is based on the fact that the *R*^2^ obtained by applying the same λ and γ value to the three propellants already exhibits a high degree of fit. However, this study found that if the requirement for the *R*^2^ is increased, by fine-tuning the λ and γ, they can be changed within a small range to meet the increased *R*^2^ requirement. There are three reasons for this small change in the λ and γ: Unavoidable measurement errors are introduced during the relaxation and dynamic modulus tests, such as errors in width and thickness measurement of the specimen, and errors in temperature conditions measured by the equipment.The time–temperature equivalent principle is used when dealing with experimental data for the relaxation modulus and dynamic modulus. This method itself is an equivalent approximation mean and also introduces errors.The overall modulus characteristics of the whole material are characterized by the individual characteristics of limited experimental specimens (3–5 specimens) rather than a large number of specimens, which also leaves considerable errors between the measured and real modulus values.

It is for these reasons that λ and γ values fluctuate within a small range, but the specific ranges of fluctuations should be: λ=1.45±0.1 and γ=0.69±0.1.

In Table 2, the optimal values for two kinds of λ and γ are given—Universal Optimal Value (λ=1.445,γ=0.691, and $R^{2}\geq 0.933$) and self-adaption optimal value (HTPB-A: λ=1.445, γ=0.691, and $R^{2}\geq 0.933$. HTPB-B: λ=1.429, γ=0.671, $R^{2}\geq 0.99$. HTPB-C: λ=1.351, γ=0.708, and $R^{2}\geq 0.99$). The universal applicability of HTPB propellants with different components is characteristic of the universal optimal value, and the self-adaption optimal value is characterized by greater accuracy.

## 5. Finite Element Simulation Analysis of HTPB Propellants

During the development of viscoelastic mechanics, its constitutive model has also developed from simple to complex.

The most basic models include the Maxwell model (Figure 7a) and the Kelvin model (Figure 7b), and then a series of more accurate constitutive models such as the viscoelastic fractional derivative model (VFD) (Figure 7c), and the integral constitutive relationship has been formed [47,48,49,50,51].

Generally, the constitutive model of a material is expressed as a generalized Maxwell model or Kelvin model composed of multiple dashpots and springs, and the mathematical expression is:(17)$$\sigma (t)+\sum_{m=1}^{M}b_{m}\frac{d^{m}}{dt^{m}}\sigma (t)=E_{0}\varepsilon (t)+\sum_{n=1}^{N}E_{n}\frac{d^{n}}{dt^{n}}\varepsilon (t).$$

Although the above equation guarantees the accuracy, it brings great difficulties to the calculation. If the fractional derivative is used in Equation (17), a single-way tension stress–strain relationship is obtained:(18)$$\sigma (t)+\sum_{m=1}^{M}b_{m}D^{\beta_{m}}\sigma (t)=E_{0}\varepsilon (t)+\sum_{n=1}^{N}E_{n}D^{\alpha_{n}}\varepsilon (t).$$

Due to the good nature of the fractional derivative, and with some reasonable simplifications, a three-parameter fractional derivative model suitable for viscoelastic materials is obtained:(19)$$\sigma (t)=E\varepsilon (t)+Et^{\alpha }D^{\alpha }\varepsilon (t).$$

Its corresponding element model is shown in Figure 7c. Since it only contains three parameters, it is commonly used in theoretical calculation.

The integral constitutive relationship is a further derivation from the Boltzmann superposition principle, which results in the Prony series expression for the relaxation modulus:(20)$$E(t)=E_{e}+\sum_{i=1}^{n}E_{i}e^{-t/\tau_{i}}.$$

Because the relaxation modulus expressed by the Prony series can be easily obtained by fitting experimental data, the constitutive model expressed by Equation (20) is commonly used in engineering testing and simulation calculation.

At present, most commercial simulation software for viscoelastic materials is obtained by fitting relaxation or creep data. However, experimental data is limited, and the constitutive model in the whole time range can only be obtained by fitting experimental data and then extrapolating it. This means that the more effective the experimental data obtained, the closer the extrapolated constitutive model is, and the more accurate the simulation can be.

For this study, the dynamic and static modulus conversion expands the time range of the relaxation modulus, and can also be considered as an extension of the master curve of the relaxation modulus. Then, a more accurate constitutive model can be obtained by fitting the extended master curve of the relaxation modulus. In this paper, to reflect the influence of different constitutive models on the simulation, the experimental curve, the master curve, and the extended master curve of the relaxation modulus for HTPB-A at 20 °C were used to define the material constitutive model for the finite element simulations. The three curves are named L^20^, L^m^, and L^m+^, respectively, in the following.

### 5.1. Fitting Analysis of the Model

Figure 8 shows the fitting results for the original data for the three models. It can be clearly seen that as the experimental data increases, the effective definition area (EDA) of the three models from small to large is L^20^ < L^m^ < L^m+^ and the fitting extrapolation area (FEA) from large to small is L^20^ > L^m^ > L^m+^. This reflects the progression of the fitted constitutive model toward the real mechanical properties of the material.

Furthermore, in the simulation software, the definition of viscoelastic material properties has to be normalized, i.e., the initial modulus of a viscoelastic material is considered as the unit “1”. The whole relaxation is the process of the modulus dropping from “1” to near “0”, which makes the initial modulus value particularly important. 

As the relaxation modulus expansion area obtained by the modulus conversion method appears in the direction close to the moment of zero load, it better reflects the mechanical properties of the propellant subjected to the instantaneous excitation. It is particularly important that the initial modulus obtained closer to the moment of zero load makes the propellant instantaneous simulation closer to the actual material situation.

### 5.2. Simulation and Its Result Analysis

The instantaneous excitation condition of HTPB propellants was analyzed by finite element simulation software. L^20^, L^m^, and L^m+^ were used to define the material properties, form the three constitutive models, and simulate the same geometric model to compare the simulation results of the three models for analysis.

Numerical simulations use a 3D model with a geometric size of 30 mm × 10 mm × 5 mm. The boundary conditions are (Figure 9a):One end is fixed, and another is loaded with a sinusoidal excitation of 50 Hz;One end is fixed and another is loaded with a constant excitation.

The results of the strain simulation are shown in Figure 9. Due to the small amount of experimental data and the low initial modulus in the L^20^ constitutive model, its strain curve fluctuations are much higher than the results of the other two curves, whereas the strain curves simulated by the L^m^ and L^m+^ constitutive models are similar. As shown in Figure 9c, the strain curve peak of L^m+^ is slightly lower than that for L^m^, and the phase difference between the two curves and the external load varies (the external loading frequency is 50 Hz and the peak occurs at 0.005 s). The difference in peaks is due to the different initial modulus and the fact that a higher initial modulus also requires a longer relaxation time when excited, so causing the phase difference.

Figure 9b shows the simulation results for the HTPB-A propellant subjected to a constant force excitation. The strain curve for L^20^ first reaches an instantaneous initial strain and then remains virtually unchanged at this level. This is because limited experimental data and a low initial modulus for the L^20^ constitutive model cause simulation results to be distorted. The strain curve for L^m^ also first reaches an instantaneous initial strain and then shows an increasing trend with an approximately constant slope. A different pattern can be seen in the L^m+^ strain curve, which shows a parabolic trend. Because of the difference in the initial modulus, the strain magnitudes at the same moment are L^20^ < L^m^ < L^m+^. However, with the increase in the loading time, the strains of all three curves will reach the same stable value.

Indeed, the difference among the three curves at the same moment in Figure 9 is obvious. This is because the higher loading frequency and shorter simulation calculation time are used to highlight the differences among the three constitutive models. If the loading frequency is reduced and the calculation time is increased, the strain curves obtained from the three models mentioned above will show almost coincident results. In other words, the contribution of the L^m+^ model proposed in this paper for simulation calculations of the propellant subjected to instantaneous excitation is outstanding, but for long-term relaxation and creep, its effect remains consistent with the existing relaxation modulus master curve model. Therefore, the study in this paper is meaningful as a useful reference for computational studies and is accurate to the millisecond level in space launches and other fields.

## 6. Conclusions


Based on the one-dimensional linear viscoelastic theory and the existing studies, the dynamic storage modulus and relaxation modulus conversion method of HTPB propellants is proposed by introducing the “catch-up factor λ” and “waiting factor γ”. The conversion method has a clear physical meaning, a simple form, and a high coefficient of determination between calculated and measured values ($R^{2}>0.93$). It is also found that for different HTPB propellants, λ and γ have the same universal optimal value.The specific values of λ and γ are determined and they show a perfect reciprocal relationship.Using this conversion method, the relaxation modulus calculated from the dynamic storage modulus can expand the time range of the relaxation modulus master curve by about four orders of magnitude. Furthermore, the dynamic storage modulus converted from the relaxation modulus can avoid the problem that the dynamic loading frequency cannot be too small.The applicability of the conversion method and introduction of λ and γ for HTPB propellants with different component contents is well verified.This method has an obvious effect on the extension of the relaxation modulus master curve. The contribution of the L^m+^ constitutive model obtained from this method is outstanding for instantaneous simulation calculations and long-term relaxation-creep simulations for propellants; the simulation results also keep consistent with existing relaxation modulus master curve models. This study has important reference value for computational studies accurate to the millisecond level in space launches and other fields.


## Figures and Tables

**Figure 1 polymers-15-00003-f001:**
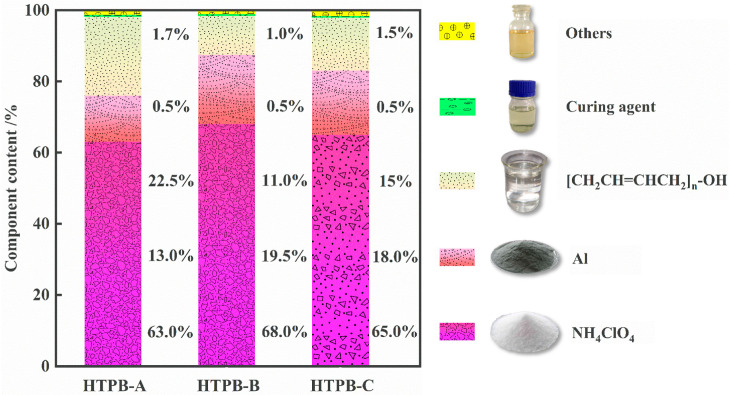
Component contents of experimental materials.

**Figure 2 polymers-15-00003-f002:**
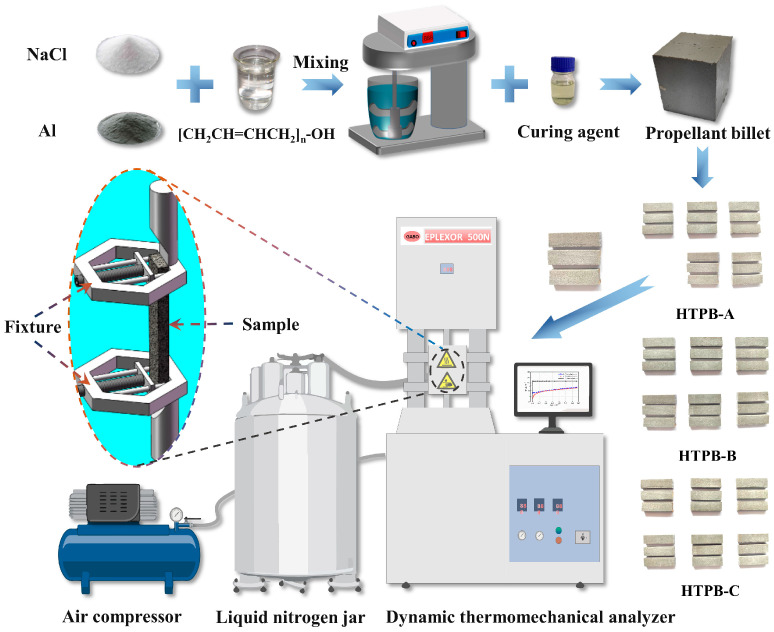
The sample preparation process and experimental equipment.

**Figure 3 polymers-15-00003-f003:**
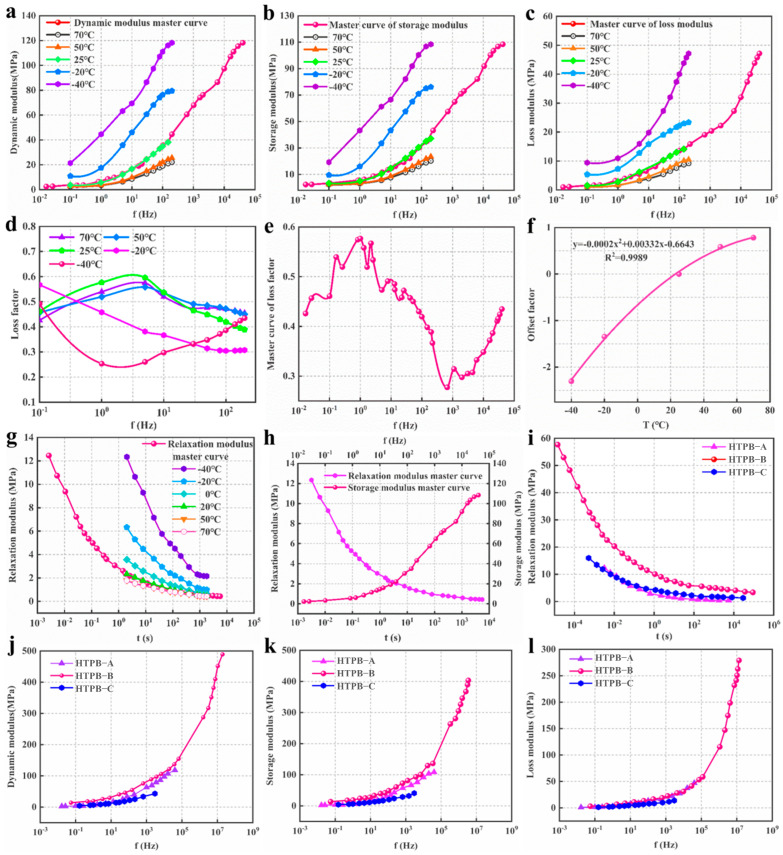
(**a**) Dynamic modulus of HTPB−A at different temperatures; (**b**) Storage modulus of HTPB−A at different temperatures; (**c**) Loss modulus of HTPB−A at different temperatures; (**d**) Loss factor of HTPB−A at different temperatures; (**e**) Loss factor master curve of HTPB−A; (**f**) Offset factor of HTPB−A; (**g**) Relaxation modulus of HTPB−A at different temperatures; (**h**) Relaxation modulus master curve and storage modulus master curve of HTPB−A; (**i**) Relaxation modulus master curve of three propellants; (**j**) Dynamic modulus master curve of three propellants; (**k**) Storage modulus master curve of three propellants; (**l**) Loss modulus master curve of three propellants.

**Figure 4 polymers-15-00003-f004:**
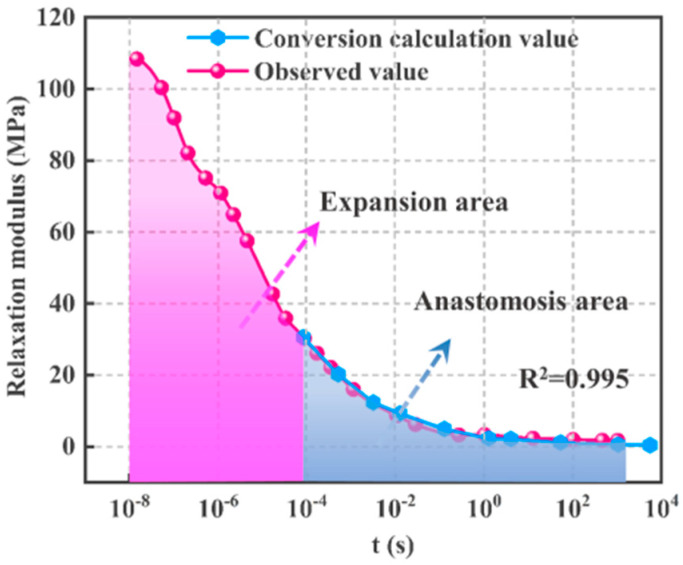
HTPB−A *E(t)*: Comparison of conversion calculated values and observed values.

**Figure 5 polymers-15-00003-f005:**
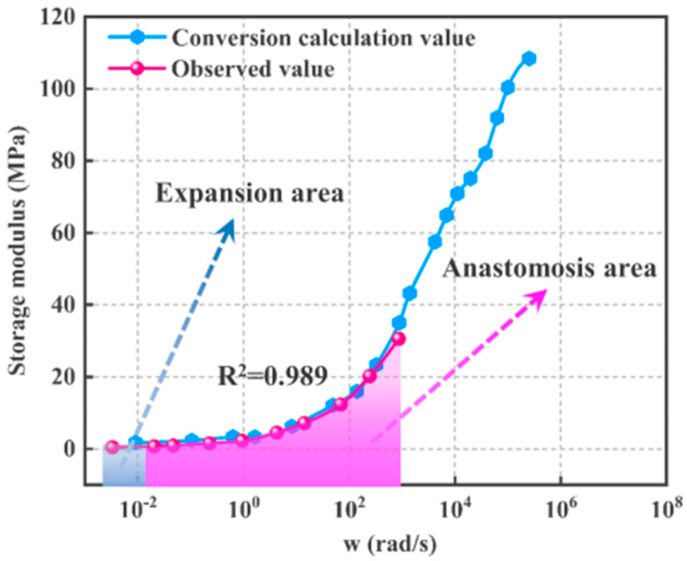
HTPB−A *E’(w)*: Comparison of conversion calculated values and observed values.

**Figure 6 polymers-15-00003-f006:**
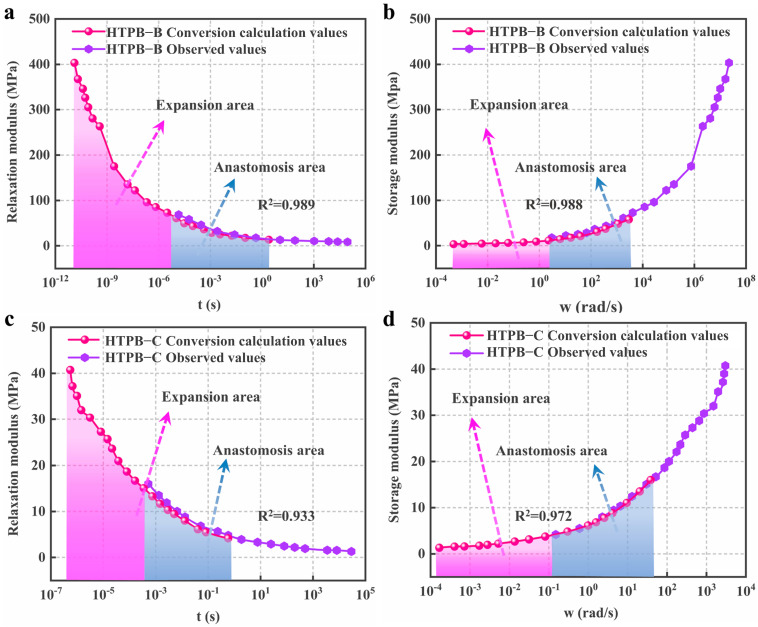
(**a**) *E(t)* calculated conversion values and observed values for HTPB−B; (**b**) *E’(ω)* calculated conversion values and observed values for HTPB−B; (**c**) *E(t)* calculated conversion values and observed values for HTPB−C; (**d**) *E’(ω)* calculated conversion values and observed values for HTPB−C.

**Figure 7 polymers-15-00003-f007:**
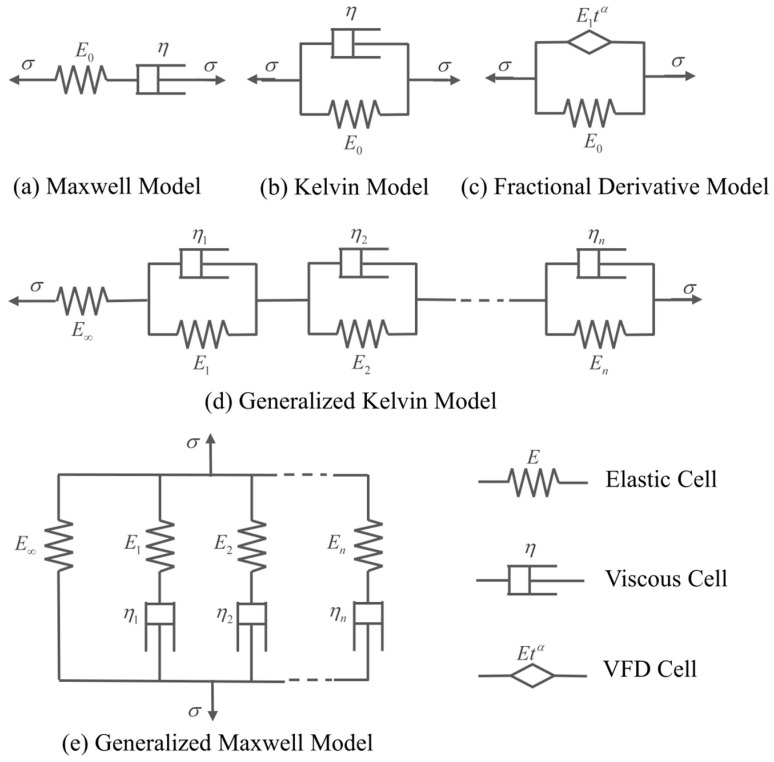
Constitutive Model.

**Figure 8 polymers-15-00003-f008:**
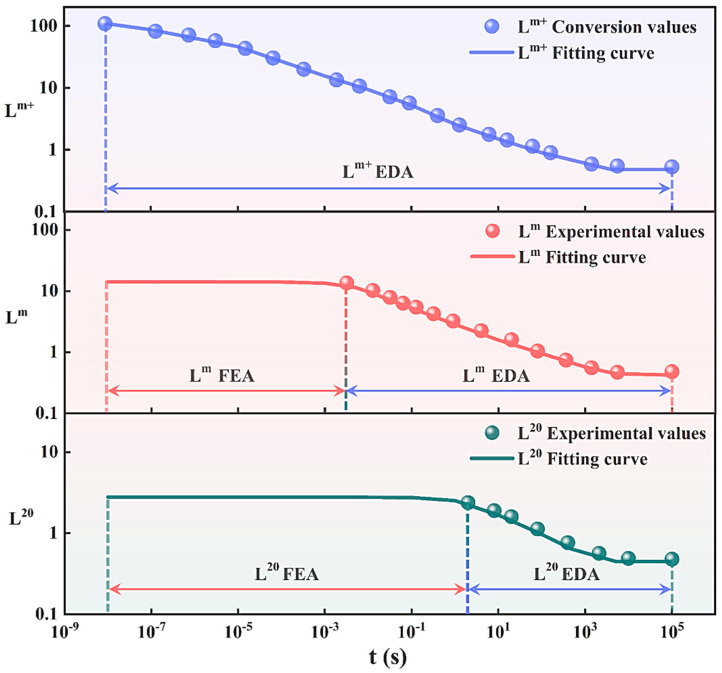
Comparison plots of the three relaxation modulus curves.

**Figure 9 polymers-15-00003-f009:**
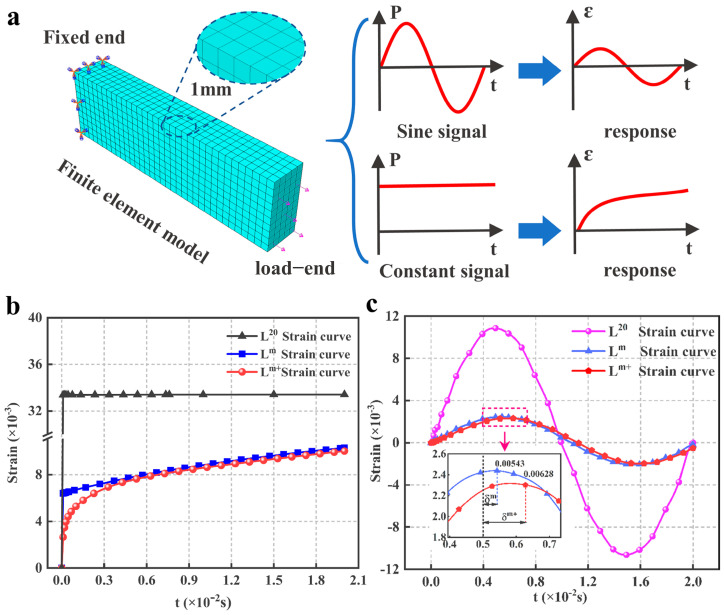
(**a**) Simulation model and input/output signals; (**b**) Constant load output signal; (**c**) Sinusoidal load output signal.

**Table 1 polymers-15-00003-t001:** Experimental condition.

Sample Type	Temperature (°C)	Relaxation Modulus Test		Dynamic Modulus Test		
		Time Range (s)	Strain	Frequency Range (Hz)	Static Strain	Dynamic Strain
HTPB-A	−20, −40, 25, 50, 70	0–1800	5%	0.1–200	5%	1%
HTPB-B	−20, −35, −55, 25, 40, 60	0–1800	5%	1–200	5%	1%
HTPB-C	−40, −20, 0, 20, 50, 70	0–1800	5%	1–200	5%	1%

**Table 2 polymers-15-00003-t002:** λ, γ, and *R*^2^ results for the modulus conversion of three propellants.

Factor	HTPB-A	HTPB-B		HTPB-C	
λ	1.445	1.445	1.429	1.445	1.351
γ	0.692	0.692	0.671	0.692	0.708
*R* ^2^	$R^{2}\geq 0.9895$	$R^{2}\geq 0.9889$	$R^{2}\geq 0.99$	$R^{2}\geq 0.933$	$R^{2}\geq 0.99$

## Data Availability

The data used to support the findings of this study are available from the corresponding author upon request.

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
