# Peer review of "Numerical Conversion Method for the Dynamic Storage Modulus and Relaxation Modulus of Hydroxy-Terminated Polybutadiene (HTPB) Propellants"

_polymers, 2022, doi:10.3390/polym15010003_

Round 1

Reviewer 1 Report

Referee Report for polymers-2088353

Journal: Polymers

Title:  Numerical conversion method for the dynamic storage modulus and relaxation modulus of hydroxy terminated polybutadi-ene (HTPB) propellants

General Comments:

The authors should address the following important points during the preparation of the revised paper

1.      The Introduction should make a compelling case for why the study is useful along with a clear statement of its novelty or originality by providing relevant information and providing answers to basic questions such as: What is already known in the open literature? What is missing (i.e., research gaps)? What needs to be done, why, and how? Clear statements of the novelty of the work should also appear briefly in the Abstract and Conclusions sections

2.      Although the mathematical concepts are correctly solved and written, some commas are missing at the end of equations. Improves the figure quality. Authors should put the detail of the Constitutive Model mentioned in figure 7 for the better understanding of the readers;

3.      Research questions are needed. Note that the results in this report are typical answers to unknown questions. This is true because the manuscript provides some powerful answers to unknown questions. Note that the research questions must connect the title to the analysis of results, and conclusion. This would guide authors not to generate many results that are not consistent to provide insight;

4.      There are few typos that are found, please review the whole article;

5.      The physics behind the conclusions is explained thoroughly but the key results should be added to the abstract;

6.      References are adequate but need careful review for punctuation;

7.      Authors should remove the typo mistakes from the whole document and also improve the language quality in the revised manuscript. Explain in detail why the shaded portion is opted in figures. For figure 6, correct the figures captions;

8.  In the abstract main findings obtained from this research paper should be added because the current form does not reflect the innovation and research significance of the paper;

9.  The following latest studies are very relevant. The authors must read and provide complete information on this topic through including these studies

a)      Finite element simulations of hybrid nano-Carreau Yasuda fluid with hall and ion slip forces over rotating heated porous cone. Scientific Reports, 11(1), pp.1-15.

b)     Numerical exploration of thermal transport in water-based nanoparticles: A computational strategy. Case Studies in Thermal Engineering, p.101334.

c)      A Galerkin strategy for tri-hybridized mixture in ethylene glycol comprising variable diffusion and thermal conductivity using non-Fourier’s theory. Nanotechnology Reviews, 11(1), pp.834-845.

Last but not least, I recommend this article for publication after providing a revised version that amends most of the raised points.

Author Response

We sincerely thank the editor and all reviewers for their valuable feedback that we have used to improve the quality of our manuscript. According to your nice suggestions, we have made extensive corrections to our previous draft, please see the attachment.

 If there are any other modifications we could make, we would like very much to modify them and we really appreciate your help.

Reviewer 2 Report

The proposed work entitled “Numerical conversion method for the dynamic storage modu- 2 lus and relaxation modulus of hydroxy terminated polybutadi- 3 ene (HTPB) propellants” is interesting and can be accepted for publication in polymers after suitable modification.

        1.        Catch-up factor l " and "waiting factor g are two terminologies are frequently used throughout the manuscript. Authors must introduce them with suitable information in the introduction section

        2.        Authors used three different propellants such as HTPB-A, HTPB-B, and HTPB-C. What is the impact of changing ammonia perchlorate ratio on overall performance?

        3.        What is the best value for l and g considering HTPB propellant?

        4    As observed from the Table 2, the deviation among l and g values are not significant.

        5.        Equations. Authors can cite them with suitable reference.

        6.        Conclusion must state the key findings of the study

Author Response

We feel great thanks for your professional review work on our article. As you are concerned, there are several problems that need to be addressed. According to your nice suggestions, we have made extensive corrections to our previous draft,Please see the attachment. 

If there are any other modifications we could make, we would like very much to modify them and we really appreciate your help.
